# Successful Pulmonary Endarterectomy after Acute Pulmonary Embolism and Reversal of Acute Cor Pulmonale in an 11-Year-Old Boy with Nephrotic Syndrome

**DOI:** 10.3390/children9101444

**Published:** 2022-09-22

**Authors:** Onyekachukwue Osakwe, Bibhuti B. Das

**Affiliations:** Department of Pediatrics, Division of Cardiology, University of Mississippi Medical Center, Jackson, MS 39216, USA

**Keywords:** acute pulmonary embolism, child, nephrotic syndrome, pulmonary endarterectomy

## Abstract

Patients with nephrotic syndrome (NS) are at an increased risk for thromboembolic events, such as deep venous and arterial thrombosis and pulmonary embolism (PE). In general, PE in children differs from adults in incidence, predisposition, pathophysiology, presenting symptoms, and management strategies. There is a lack of treatment guidelines for PE in children, and the management strategies are mostly extrapolated from adult data. This case report highlights the presentation of acute cor pulmonale due to massive PE associated with NS and a successful pulmonary endarterectomy that reversed the child’s pulmonary hypertension and normalized right ventricular function.

## 1. Introduction

Patients with nephrotic syndrome (NS) are at an increased risk for thromboembolic events, such as deep venous thrombosis, arterial thrombosis, and pulmonary embolism (PE). This thromboembolism phenomenon has been attributed to a “hypercoagulable” state because of a combination of proteinuria, hypercholesterolemia, and hypoalbuminemia correlated with endogenous thrombin-induced hypercoagulopathy [1]. However, unlike for adults with PE, the risk factor assessment and management of PE among children with NS is insufficiently studied [2,3]. Zhu et al. identified proteinuria and membranous nephropathy as independent risk factors for developing PE in children with NS [3]. The epidemiology of NS-associated thromboembolism differs significantly between children and adults [4]. The computed tomography pulmonary angiography (CTPA) technique identified the incidence of PE associated with NS in children and young adults as being as high as 25% [3]. These findings support the need for more effective thromboembolism prophylaxis in children with NS.

The classic triad of PE symptoms consists of pleuritic chest pain, shortness of breath, and hemoptysis, which is common in adults but is rare in children. Hypoxemia and dyspnea are common in children after acute PE. The diagnosis of PE can be missed in children because PE symptoms are nonspecific and mimic other common childhood conditions, such as pneumonia. The determination of the measurements of antithrombin III, fibrinogen, and D-dimer help diagnose PE in adults but are not specific in children [5]. A CTPA is the first line of testing in children with suspected PE, especially with thromboembolic risk factors, such as NS. In adults, current guidelines for the management of PE recommend antithrombotic and thrombolytic therapy for acute PE. However, management strategies for acute PE vary in children due to a lack of standard guidelines. In adults, pulmonary artery-catheter-directed thrombolysis and catheter embolectomy are potentially effective therapeutic options [6]. Catheter-directed thrombolysis has been described and is feasible in children, but the experience is limited [7]. Catheter-derived thrombolysis in children may result in acute hemorrhagic strokes and more studies are needed to assess safety and effectiveness. We describe an 11-year-old boy with NS who had acute cor pulmonale due to massive PE, did not respond to continuous heparin and activated tissue plasminogen activator infusions, but subsequently had a rapid resolution after a successful pulmonary endarterectomy (PEA). We discuss the feasibility of PEA and review the literature on the appropriate use of PEA in children with acute cor pulmonale due to PE.

## 2. Case Report

An 11-year-old Caucasian boy with a history of chronic steroid-resistant NS secondary to minimal change disease was admitted to the nephrology service at our hospital. He was on tacrolimus, prednisone, enalapril, and multiple oral diuretics. He developed sudden onset worsening hypoxic respiratory failure with increasing edema and ascites. His initial chest x-ray showed worsening pulmonary edema. His dyspnea improved with fluid removal after initiating hemodialysis, albumin infusion, and high-dose Solu-Medrol, but his oxygen saturation remained at 70%, and he continued to require supplemental oxygen. His blood workup showed elevated D-dimer (53, 317 ng/mL), elevated fibrinogen (554 mg/dL), decreased albumin (2 g/L), elevated BUN 62 (mg/dL), and elevated creatinine (4 mg/dL). He had normal PT (11.1 s), normal APTT (34.95 s), elevated antithrombin III (131%), increased protein C (159%), increased protein S (154%), a negative value for Factor V Leiden variant c.1601 (G > A; p. Arg534Gln), no prothrombin G20210A, and no antiphospholipid antibody. However, he had a very high level of cardiolipin IgM antibody (56 MPL, normal < 12 MPL). A 12-lead ECG showed an “S1Q3T3” pattern of acute cor pulmonale (Figure 1).

His echocardiogram revealed thrombi in the right atrium (RA) (Figure 2A) and right pulmonary artery (RPA) (Figure 2B), a dilated right ventricle (RV) with decreased function (Figure 2C), and supra systemic RV pressure (Figure 2D). He underwent a ventilation/perfusion (VQ) scan, which revealed perfusion defects involving the entire right lung, and lower left lung (Figure 2E). A CTPA confirmed complete occlusion of RPA and non-occlusive thrombi in the left lower branch pulmonary artery (Figure 2F). He was treated with continuous heparin and activated tissue plasminogen activator infusions, but there was no significant improvement. Catheter-directed therapy was attempted to remove the RPA thrombus but was unsuccessful. The procedure comprised of RA thrombectomy using a 20 French Inari FlowTriever thrombectomy catheter to serially aspirate large amounts of chronic, well-organized thrombi. However, the RA thrombus could not be completely retrieved. This was followed by serial attempts at the removal of the RPA thrombus with a Triever 24 catheter, followed by a FlowTriever 2 disk catheter. At the beginning of the procedure, his mean PA pressure was 40 mmHg with 100% oxygen and 20-ppm nitric oxide (Figure 2G), but he quickly developed a pulmonary hypertension (PH) crisis during catheter manipulation. The procedure was aborted, and he subsequently underwent PEA and the removal of the RA thrombus (Figure 2H). Immediately after the procedure, his RV pressure normalized (Figure 2I). A repeat VQ scan showed improved perfusion in the right and left lungs (Figure 2J), and a repeat CTPA showed the complete resolution of the thrombus in RPA and no new emboli (Figure 2K). Three months after PEA, his echocardiogram showed significant RV function improvement and no evidence of PH recurrence. He continues to require renal supplemental therapy and anticoagulant prophylaxis with low-molecular-weight heparin (LMWH), as he had other complex medical issues. He had an episode of intrabdominal hemorrhagic complications and required the exploration of his abdomen to stop the bleeding. We plan to change to direct oral anticoagulants (DOAC), such as rivaroxaban, in future, which was recently approved by the US Food and Drug Administration (FDA) in children based on two randomized control trial [8,9]. The reason to continue with LMWH in our case was because of the poor renal function and uncertain clearance of rivaroxaban by hemodialysis.

## 3. Discussion

Acute PE is a rare but known complication in the setting of hypercoagulation associated with NS. When there is sudden onset hypoxemia, as in our case, there should be a high index of suspicion for PE, which is needed to establish an early diagnosis and provide immediate treatment, in order to allow for a favorable outcome. The patient described herein had a massive PE during his in-hospital treatment for steroid resistant NS and CTPA established a definitive diagnosis of his acute PE. In this patient, PE was associated with acute pulmonary hypertension and acute severe RV dysfunction leading to acute cor pulmonale. There is no preferred method of treatment for acute PE with hemodynamic compromise in an 11-year-old child, and we discuss our clinical experience in this case supported by evidence from the literature.

A pulmonary angiogram is the gold standard for diagnosing acute PE. It is an invasive procedure with associated risks of sedation/anesthesia, arrhythmia, bleeding, infection, and pulmonary hypertensive crisis. It has been replaced with less invasive methods, such as CTPA, which is now the primary modality for assessing PE. The partial or complete pulmonary arterial filling defect is usually suggestive of acute PE. A V/Q scan has been considered helpful for diagnosing PE, and it is safe, sensitive, reproducible, and uses relatively low radiation without the need for iodinated contrast media. However, the V/Q mismatch is not specific to PE. It can be falsely positive in several other conditions, including congenital and acquired pulmonary arterial stenosis, air, fat, and foreign body embolism, and pneumonia. The evaluation of the heart by echocardiogram is essential in the setting of acute PE because evidence of RV dysfunction is correlated with higher mortality [10].

Prevention and management strategies for acute PE vary in children due to a lack of standard guidelines. The updated American Society of Hematology (ASH) 2020 guidelines for the management of venous thromboembolism in adults with PE and hemodynamic compromise recommend using thrombolytic therapy followed by anticoagulation over anticoagulation alone for acute PE [11]. In our patient, there was echocardiographic evidence of acute RV dysfunction due to massive PE and no improvement after use of heparin and, subsequently, after using systemic thrombolytic therapy. When antithrombotic and thrombolytic therapies are ineffective in resolving thrombosis, there is an option to use pulmonary artery catheter-directed therapy. However, recent ASH guidelines in adults with PE recommend that systemic thrombolysis is preferred over catheter-directed thrombolysis because of the uncertainty about the outcomes due to the lack of randomized trial data and variability in procedural experience across centers [11]. Furthermore, there is a paucity of data on the safety and efficacy of catheter-directed therapy for PE in children.

In our case, the catheter-derived removal of the thrombus was ineffective, and we used surgical PEA to improve RV dysfunction, decrease hypoxia by improving lung perfusion, reverse acute pulmonary hypertension, improve RV function and subsequent associated complications. One of the major complications after acute PE is chronic thromboembolic pulmonary hypertension (CTPH), which occurs in 3.2% of acute PE survivors [12]. To our knowledge, this is the first case of PEA in a child developing acute cor pulmonale due to massive PE associated with NS. Prior published reports have demonstrated that PEA is feasible and well tolerated to relieve acute PE secondary to different comorbidities [13,14,15,16,17,18,19]. Madani et al. have shown that after PEA, there was a significant improvement in hemodynamics, RV function, and the functional status of patients with minimal postoperative complications and low perioperative mortality, similar to that reported for adults with CTEPH, with the notable exception being a higher rate of re-thrombosis in pediatric patients [19]. However, PEA is not a risk-free procedure and can be associated with increased mortality [20].

Following the completion of the treatment of acute PE, secondary prophylactic anticoagulants are needed for at least six months or even indefinitely until the NS is in remission or definite renal replacement is performed. The FDA has recently approved the use of rivaroxaban (DOAC) in children, especially after Fontan surgery for single ventricle congenital heart disease; however, there remain a number of unanswered questions on using DOAC in children. In children with acute venous thromboembolism, treatment with rivaroxaban resulted in a similar low recurrence risk and reduced thrombotic burden without increased bleeding, compared with standard anticoagulants [21]. The usage of rivaroxaban could overcome the limitation of warfarin or LMWH (mainly the necessity of regular laboratory monitoring and parenteral application) while providing similar efficacy and safety to treat venous thromboembolism in children [8]. Anticoagulant therapy is very effective at preventing recurrent PE but is associated with an increased frequency of bleeding complications, although relatively lower risk for major bleeding with DOAC.

## 4. Conclusions

This rare case highlights a pediatric patient who underwent PEA for acute cor pulmonale due to acute massive PE associated with NS. The successful management of our patient by PEA required a multi-disciplinary team, including cardiac surgeons, hematologists, cardiac intensivists, and cardiologists. Furthermore, this case, along with other cases reported in the literature [13,14,15,16,17,18,19], demonstrate that young age is not a contraindication for PEA, and this may be the single most effective treatment for acute cor pulmonale due to massive PE when the thrombus is not resolved with standard anticoagulation and anti-thrombolytic therapy.

## Figures and Tables

**Figure 1 children-09-01444-f001:**
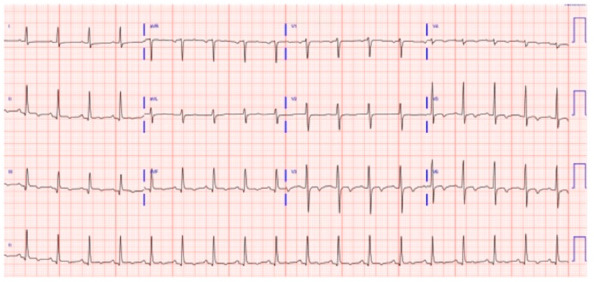
Twelve-lead Electrocardiogram showing sinus tachycardia and “S1Q3T3” pattern of acute cor pulmonale. Note S wave in lead I, a Q wave in lead III, and an inverted T wave in lead III indicate acute right heart strain.

**Figure 2 children-09-01444-f002:**
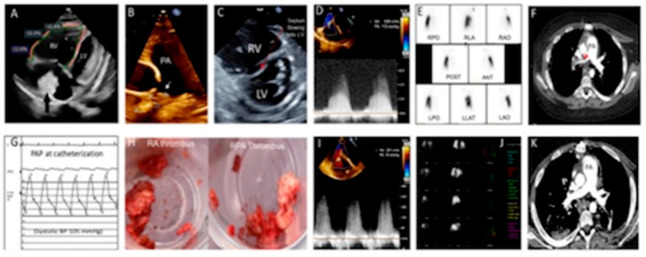
(**A**) Four-chamber view showing dilated RV, decreased RV global longitudinal strain, RVEF (43%) by 3-D, and a large thrombus (measuring 34 mm × 26 mm) in RA; (**B**) 2-D echocardiogram showing mild dilated pulmonary artery (26 mm) and a large thrombus measuring 26 mm × 12 mm occluding RPA; (**C**) parasternal short-axis view showing dilated RV with inter-ventricular septum bowing into LV in systole suggesting supra-systemic RV pressure; (**D**) TEE prior to surgery showing RV systolic pressure 112 mmHg plus RA pressure (systemic systolic BP was 105 mmHg); (**E**) ventilation-perfusion (V–P) scan prior to pulmonary artery endarterectomy (PAE) showed large unmatched perfusion defect in entire right lung and decreased perfusion of mid lower left lung; (**F**) computed tomography pulmonary angiography (CTPA) showed a large thrombus in RPA, occluding blood flow to right lung; (**G**) at right heart cardiac catheterization while on 100% oxygen and iNO, PAP was 58/29, mean 40 mmHg; (**H**) thrombus removed from RPA and RA; (**I**) immediately after PAE RV, systolic pressure based on TR velocity obtained by TEE was significantly decreased to 23 mmHg plus RA pressure; (**J**) post-PAE repeat VP scan showed improved perfusion defects in right lung and mid lower left lung; (**K**) post-PAE repeat CTPA showed no thrombus in RPA and no new emboli.

## Data Availability

Not applicable.

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
