# Peer review of "Successful Pulmonary Endarterectomy after Acute Pulmonary Embolism and Reversal of Acute Cor Pulmonale in an 11-Year-Old Boy with Nephrotic Syndrome"

_children, 2022, doi:10.3390/children9101444_

Round 1

Reviewer 1 Report

The authors describe the rare occurrence of a pulmonary embolism in a pediatric patient with nephrotic syndrome. This is a novel topic, and one that is important to publish, given the rare occurrence of the events, leading to possible difficulty in diagnosis and management. 

The introduction seems to focus on the diagnosis of a pulmonary embolism, so it was somewhat surprising that the last few lines mention that the authors' focus would be on describing PEA management. It may be beneficial to provide more focus on PEA versus medical or cath management in the introduction. 

The case itself and the results were presented well. The details were clearly conveyed, and overall the graphics presented added to the story of the case. The only figure that I had trouble seeing was the VQ scans. Not sure if there may be room in the figure to label right versus left lung. 

Overall good discussion of various management strategies of a pulmonary embolism in a pediatric patient. I would be interested to have the authors expand more on the risks of a PEA. It is somewhat misleading to lead the discussion with comparison of diagnostic techniques. An introductory statement outlining the focus of the discussion may help to better organize this part of the manuscript. 

Author Response

Response to Reviewer-1:

The authors describe the rare occurrence of a pulmonary embolism in a pediatric patient with nephrotic syndrome. This is a novel topic, and one that is important to publish, given the rare occurrence of the events, leading to possible difficulty in diagnosis and management. 

The introduction seems to focus on the diagnosis of a pulmonary embolism, so it was somewhat surprising that the last few lines mention that the authors' focus would be on describing PEA management. It may be beneficial to provide more focus on PEA versus medical or cath management in the introduction. 

- Thank you. We edited and changed the introduction to incorporate medical vs. catheter-directed therapy for PE.

The case itself and the results were presented well. The details were clearly conveyed, and overall the graphics presented added to the story of the case. The only figure that I had trouble seeing was the VQ scans. Not sure if there may be room in the figure to label right versus left lung. 

- Thank you. We labeled the figure depicting the V/Q scan with Right vs. Left Lung with appropriate views in revised Figure-2

Overall good discussion of various management strategies of a pulmonary embolism in a pediatric patient. I would be interested to have the authors expand more on the risks of a PEA. It is somewhat misleading to lead the discussion with comparison of diagnostic techniques. An introductory statement outlining the focus of the discussion may help to better organize this part of the manuscript. 

- Thank you. We have edited the discussion section to incorporate more on the risk of PE. We added that at the beginning of the discussion and rearranged the discussion section.

Reviewer 2 Report

This is a very educative case report alluding to a complication that is not that rare in a patient cohort with nephrotic syndrome.

Most of the diagnostic work up is comprehensive. As more and more patients with PE are treated by transcatheter measures the authors should add which trans catheter technique that finally failed, has been used.

From the pictures the pressure tracing curve could be removed, as it would only add further information if the systemic pressure curve  would have been depicted in the same figure. Just citing the numbers is also very convincing that the pressures were elevated.

The authors do not mention which medication was used for anticoagulation after thrombectomy - this should be added.

Would the authors like to make a recommendation for patients with nephrotic syndrome for screening for pulmonary embolism, as they are at continuing risk?

Author Response

Response to reviewer-2

This is a very educative case report alluding to a complication that is not that rare in a patient cohort with nephrotic syndrome.

- Thank you.

Most of the diagnostic work up is comprehensive. As more and more patients with PE are treated by transcatheter measures the authors should add which trans catheter technique that finally failed, has been used.

- We discussed catheter-directed management in our case report section in detail.

From the pictures the pressure tracing curve could be removed, as it would only add further information if the systemic pressure curve would have been depicted in the same figure. Just citing the numbers is also very convincing that the pressures were elevated.

- Thank you. We added the systemic pressure to the Figure depicting the PA pressure.

The authors do not mention which medication was used for anticoagulation after thrombectomy - this should be added.

- We added that our patient remained on LMWH for secondary prophylaxis as he had bleeding complications, but our preference is to start rivaroxaban for anticoagulation therapy.

Would the authors like to make a recommendation for patients with nephrotic syndrome for screening for pulmonary embolism, as they are at continuing risk?

- We added that PE is a risk factor in the setting of hypercoagulation associated with nephrotic syndrome. We want to emphasize that when there is sudden onset hypoxemia as in our case, one should have a high index of suspicion for PEA.